# *"I wish they heard my story rather than my conditions."* –A qualitative exploration of young people's experiences during mental health assessment in the UK

**Alessandro Dorata**[1]*, **Hannah Andersen**[2], **Sarah Bisp**[3], **Rebecca Appleton**[4]

1 Division of Psychiatry, University College London, London, United Kingdom, 2 Division of Psychiatry, University College London, London, United Kingdom, 3 Department of Nursing, Midwifery & Health, Northumbria University, London, United Kingdom, 4 Department of Epidemiology and Applied Clinical Research, University College London, London, United Kingdom

* alessandro.dorata.22@alumni.ucl.ac.uk

## Abstract

Poor mental health can present a significant impact on young people's quality of life. Mental health assessment may detect issues early and select appropriate treatments to prevent a worsening of symptoms. However, current research suggests that rates of non-attendance for mental health services are high amongst young people, possibly due to negative assessment experiences. Despite their potential importance, little research has explored the experiences of young people during mental health assessments. We recruited 12 culturally diverse young people who have had a mental health assessment within the last 18 months. They participated in semi-structured online interviews exploring their experiences during assessment. We analysed the data using reflexive thematic analysis. Young people with lived experience helped identify the themes for data analysis. Three themes were identified: *importance of person-centred care, systematic barriers* and *safe space*. Overall, participants reported an unfulfilled desire for holistic and personalized care that prioritises their needs over meeting systematic requirements. Assessment was frequently characterised by a lack of agency and dismissal of experiences, generating disengagement. Young people who reported a more positive experience of assessment identified factors including a warm environment that facilitated engagement and prepared participants for upcoming support. The results suggest that, although mental health assessments have the potential to detect mental health issues early and facilitate subsequent treatment engagement, young people often experience them negatively. Frequently, they are perceived as impersonal and rigid, presenting a barrier to help-seeking. In light of these issues, there is a need to prioritise the implementation of person-centred care in assessment practices.

**Data availability statement:** In compliance with our ethical approval from University College London's Research Ethics Committee (Reference Number: 27347/001), individual transcripts from this study cannot be publicly shared. Participants did not consent to the disclosure of these transcripts, and they contain potentially identifying or sensitive information, meaning there is a risk that public access would compromise patient confidentiality. Data access requests can be directed to the corresponding author and to UCL's ethics committee ethics@ucl.ac.uk.

**Funding:** As this is a master's dissertation, this project did not receive grant funding from an external funding body. This study was supported by UCL's Division of Psychiatry in the form a small amount of money to recruit and compensate participants as well as PPIE. This money was available to all projects within this cohort and not specifically awarded to this project. No salary or other financial contributions were made to AD or the researchers involved. The specific roles of this author are articulated in the 'author contributions' section. UCL had no role in study design, data collection and analysis, decision to publish, or preparation of the manuscript.

**Competing interests:** The authors have declared that no competing interests exist.

## Introduction

Mental health issues among young people between the ages of 10–24 are a growing global concern, with a significant number of individuals in this age group experiencing anxiety, depression, and other mental health conditions [1]. According to the World Health Organization (WHO), mental health conditions are the leading cause of disability among 10–24-year-olds worldwide [2]. In addition, mental health issues are among the leading causes of disability-adjusted life years (DALYs) lost among young adults [3], demonstrating how, if left unanswered, mental health issues can have profound impacts on young people's quality of life and social functioning.

Young people are especially susceptible to mental health issues as they are navigating a period characterised by significant social, psychological and biological transitions [4,5]. Although turbulent, this period may present a special opportunity for support before any issues escalate and become chronic [6],

Thus, early detection and intervention are crucial in preventing the progression of these issues and mitigating any long-term impact [7]. Given the myriads of factors contributing to mental health issues, especially in young people, any such detection ought to account for them holistically [8,9]. However, despite the acknowledged importance of mental health care for young adults, there is a notable scarcity of research exploring their experiences during mental health assessments, which is a critical first step in the therapeutic process, inaugurating people to the mental health system [10].

Mental health assessments may incorporate interpersonal interview components or standardised questionnaires aimed at facilitating patient self-disclosure [11] to reveal symptom patterns and help ascertain service eligibility and treatment fit [12]. Across domains, mental health assessments appear to be heterogeneous and non-standardised, whereby different service providers, i.e., primary or secondary care may follow different, service specific protocols [13]. There is some evidence that suggests that the very act of self-disclosure in therapeutic settings may be important in determining the subsequent therapeutic trajectory and outcomes of therapy by indexing patients' engagement [14]. However, young people may have difficulties divulging sensitive information in standardised ways due to perceived stigma or mistrust in the system [15,16]. Furthermore, recent qualitative evidence posits that young people report struggles engaging with assessments that are primarily held over the phone [17]. It is thus key to determine age-appropriate ways of promoting self-disclosure and engagement during assessments. Despite assessments' potential importance, very little research has been done investigating the experience during their mental health assessments and how young people make sense of that experience. Most of the extant literature scrutinises assessment through the lens of psychometrics and primarily focuses on to the validity and reliability of diagnostic tools [18,19]. Although mental health services routinely deploy validated psychometric tools, such as the PHQ-9 or the GAD-7 [20,21] during assessments, they do not exclusively rely on the usage of these tools [22] and frequently involve interpersonal interview components. Thus, research that prioritises the validation of psychometrics may obscure important

parts of the assessment process. Although there has been some recognition of culturally driven mental health disparities, there has been a lack of research scrutinising the ways in which assessments are experienced emotionally and how cultural differences may play an important role in altering that emotional experience [22].

Moreover, evidence suggests that attrition from mental health services starts well before the start of any treatment, alluding to assessments' pivotal role in maintaining engagement. In 2014, for example, more than half of the patients who had undergone assessment within NHS Talking Therapies in England did not attend their first treatment session [23]. Sweetman et al. [24] suggest some risk factors for disengagement, such as referral pathway, young age, male gender and minoritised ethnicity. Other evidence attributes attrition to assessment mismanagement due to practitioner biases, affecting predominantly marginalised people [25]. Further research shows that feeling misunderstood or stigmatised can deter young adults from seeking any further help [26]. However, none of these analyses provide any insight into the experiences during assessment that might increase rates of nonattendance.

Given the dearth of research on this subject and its potential importance to treatment outcomes and attendance, the primary aim of this study is to explore the experience of young adults during mental health assessments and attain an understanding of the unique challenges faced by that population.

## Methods

These findings have been reported with reference to the COREQ guidelines for reporting qualitative research [27].

We conducted online qualitative semi-structured interviews to explore young people's experiences of mental health assessments within the UK.

### Ethical considerations

This project received ethical approval from the UCL Research Ethics Committee prior to data collection (Reference Number: 27347/001).

### Participants and recruitment

The target population was young adults between the ages of 18–26 [28] who had a mental health assessment in the UK within the previous 18 months at the time of recruitment. This study aimed to recruit 10–12 participants, reconciling temporal constraints with maximising a demographically diverse sample. As this study was exploratory, participants were not stratified according to assessment modalities. Participant recruitment started 01/05/2024 and ended 20/06/2024. We maximised the sample diversity by using purposive sampling [29] and captured the voices of people who have historically not always received empirical representation [30]. As such, this project specifically looked to recruit people from different ethnic backgrounds, including Black and Asian backgrounds, various sexual orientations, as well as men. We recruited the initial seven participants via the mailing lists of institutions within the voluntary sector, namely, McPin, NSUN and The Mix. Potential participants were asked to email the study team to register their interest. Once potential participants had gotten in touch, they were asked to provide some demographic data (age, gender, and time since last assessment) to ensure they fit the inclusion criteria and allow for a diverse sample. After initial recruitment, the sample consisted of only two men (28.6%) and no people from Black ethnic backgrounds. To increase participants from these groups, a recruitment poster was shared on the social media platform X, resulting in further recruitment of five participants, four of which were Black men. To ensure that participants did not give fraudulent profiles or were using AI to get in touch [31], before the interview, they were asked to confirm their demographic data again and briefly turn on their webcam to check their legitimacy. We did not record these interactions to ensure participant anonymity.

### Data collection

Eligible participants were sent a copy of the consent form and participant information sheet via email and were offered the opportunity to ask questions. Participants gave verbal informed consent prior to the interview over Microsoft Teams which

was recorded separately by the interviewer immediately before the interview began. Audio recordings of consent were securely stored on a password-protected, access-controlled shared drive. The UCL Research Ethics Committee reviewed and approved the consent procedure, including the use of verbal consent. Interviews were all conducted by AD via Microsoft Teams. Participants were given a £25 voucher to thank them for their time. We collaborated with young people with lived experience to create a pre-determined topic guide (S1 File), which framed the interviews. The topic guide was chronologically structured and included questions regarding participants' referral process, assessment experiences and post-assessment feelings. Fig 1 shows some example questions.

## Data analysis

We recorded all interviews using Microsoft Teams. Recordings were transcribed verbatim and checked for accuracy before being permanently deleted. All identifiable information was removed during transcription.

Having imported the edited transcripts to NVivo v.14, we analysed them using Braun and Clarke's [32] framework for reflexive thematic analysis. As such, no predetermined theories or themes were applied to the data and analysis was inductive. Initially, AD familiarised himself with all transcripts and then generated preliminary inductive codes. Both process coding as well as descriptive coding was used to generate these codes [33]. The former aided in capturing actions and feelings regarding assessment, whereas the latter was used to summarise broader topics in single descriptive codes. To sense-check ideas and explore multiple perspectives of the conceived coding framework [34], another researcher (HA) independently coded 25% of the transcripts. Afterwards, the framework was adjusted according to any new insights reported by the second coder. Subsequently, the framework was iteratively calibrated by further collating or expanding existing codes and creating appropriate themes or subthemes. Where possible, process codes were allocated to descriptive codes to form preliminary themes, associating actions and feelings with the general descriptor of the theme. We ensured fit with the data by applying the resulting thematic framework to four representative transcripts before completing data analysis on all remaining transcripts. All members agreed on the themes before full data analysis.

## PPI involvement

Following Jennings et al. [35] framework for best practice for patient and public involvement in qualitative research, two young people with lived experiences of using mental health services were involved with the creation of the topic guide and the analysis of the data. They were members of an existing lived experience advisory group that had previously collaborated with a member of the research team (RA) on a separate project, and were invited to contribute to this study based on their relevant expertise and established working relationship. Two separate online PPI panels were held. In the first panel, contributors suggested revisions to the topic guide to improve clarity and relevance, particularly around language,

Interview on the experience of Mental Health Assessment

**Feelings prior to assessment**
- What were your initial expectations of the assessment before attending?

**Experience during the assessment**
- What was important to you to get across during the assessment?
- During the assessment, did you feel that your experience was heard and understood?

**Experience post assessment**
- How did you feel once the assessment was over?
- Did you feel as though your assessment experience had any impact on your treatment?

**Fig 1. Example interview questions.**

tone, and the framing of sensitive topics. In the second panel, they reviewed initial thematic summaries and helped shape the preliminary analytic framework. Their insights informed the final coding structure and interpretation of findings.

**Reflexivity & data validity**

The primary researcher of this project, AD, a white male postgraduate student, has had roughly two years of experience working in mental health care, specifically in Talking

Therapies, as a psychological wellbeing practitioner, assessing and treating people with mental health issues. RA is a female researcher with expertise in young people's mental health and mental health policy, whose PhD work inspired the current project's trajectory. SB is a female assistant professor in mental health nursing. HA is a female postgraduate student with a keen interest in mental health research.

Considering the primary researcher's clinical background, there were preconceived notions as to what mental health assessment may look like. For example, Talking Therapies has a very standardised way of conducting assessments. Thus, the questions pertaining to the draft topic guide were designed with that experience in mind and may have initially been too myopic. After PPI consultation, some amendments were made to accommodate a broader spectrum of assessments. This prior experience may have also influenced interview prompts to be congruent with preconceived notions of assessment. On the other hand, first-hand experience of assessments may have allowed the primary researcher to be more attuned to participants' experiences drawing out more in-depth answers.

Being familiar with people's anecdotally negative experiences of may have also influenced the interviews to reflect a more negative point of view. The lead researcher tried to account for this bias in the topic guide by incorporating a section on positive assessment experiences. This issue was also openly discussed with this project's PI to remain honest about expectations and biases.

AD's educated and ethnically privileged background may have introduced power imbalances during the interview, which could have discouraged participants from sharing culturally sensitive experiences. Furthermore, as a white man, AD may not be privy to the nuances of marginalisation, which may have been absent from the topic guide or follow-up question during the interviews.

To remain reflexive about their own biases, the primary researcher also kept a reflexive diary during the research process to minimise the impact of any biases.

## Results

We recruited and interviewed twelve participants. Participants were between 18 and 26 years old with a mean age of 22.5 years (SD = 2.61). Five participants identified as male (41.6%) whilst another five participants identified as female (41.6%). The remaining two participants identified as non-binary (16.7%). Participants had a variety of self-identified sexual orientations and were from different ethnic backgrounds (Table 1). Interviews lasted between 31 and 61 minutes with a mean length of 44 minutes (SD = 11.5).

Three themes were identified that capture participants' mental health assessment experiences, each consisting of several subthemes (Fig 2). The primary themes are *importance of person-centred care*, *systematic barriers* and *safe space*. Since there was no participant stratification according to different assessment modalities, we did not pursue themes that explored the specifics of a single modality as this was outside of this project's remit. The identified themes were recurrent, rich and coherent, directly relating to the central aim of this study. This can be understood as different assessment modalities eliciting similar core experiences that warrant an analysis of assessment experiences as a whole.

### Theme 1: Importance of Person-centred care

Although this is a standalone theme, it also pervades and contours every other theme, highlighting participants' fundamental desire to be heard and understood for who they are as a person beyond their apparent mental health problem(s).

**Table 1. Breakdown of demographic data of study participants.**

| Demographics | Number of Participants |
|---|---|
| **Mean Age in Years (Standard Deviation)** | 22.5 (2.61) |
| **Age Range in Years** | 18-26 |
| **Mean Time Since Last Assessment in Months (Standard Deviation)** | 5.25 (4.84) |
| **Gender** | |
| Men | 5 (41.6%) |
| Women (Including 1 Trans-Woman) | 5 (41.6%) |
| Non-Binary | 2 (16.7%) |
| **Sexual Orientation** | |
| Heterosexual | 2 (16.7%) |
| Bisexual | 4 (33.3%) |
| Homosexual | 3 (25%) |
| Asexual | 2 (16.7%) |
| Undecided | 1 (8.3%) |
| **Ethnicity** | |
| White/White British | 4 (33.3%) |
| South Asian/Asian British | 3 (25%) |
| Black Caribbean/ Black British | 5 (41.6%) |

Person-centred care has been divided into *patient and person dichotomy*, *lack of agency*, *psychometric tools as hurdles to being understood* and *diversity*.

**Patient and person dichotomy.** Participants consistently brought up a dichotomous experience of being treated as a patient rather than a person. Being treated as a patient emphasised medicalised views of mental health, making their conditions more salient than their personal stories and pathologizing their lives.

*"I think, for me, it's a simple difference in that my mental health problems are one aspect of me and one part of my life that sometimes causes some issues...Am I a young person who's doing a lot of things and sometimes I also get depressed…, or am I just a depressive person?" (YP002, female)*

Being treated as a patient is often seen unfavourably and entails the quantification of people's symptoms. In these cases, the majority of participants did not feel listened to or understood, and their experiences were seemingly coopted by preconceived notions of mental health.

*"I feel like a patient is just judged on those numbers and scores that you give at the start of every session... Whereas a person is actually, it takes the whole thing into account, and you're treated like an actual individual and you're listened to." (YP007, male)*

Participants often felt that being a patient felt transactional and impersonal, reminiscent of being at a "bank and setting up a bank account". Consequently, participants did not feel as though their issues were taken seriously, negatively impacting their ability to engage. Assessments where participants were treated as patients, opposed to people, were characterised by power imbalances that elevated the status of the assessor to one that "knows all", disempowering the person who was assessed.

*"…they always see me as a patient. It's very much an experience of, "I know better than you about your own experiences."" (YP002, female)*

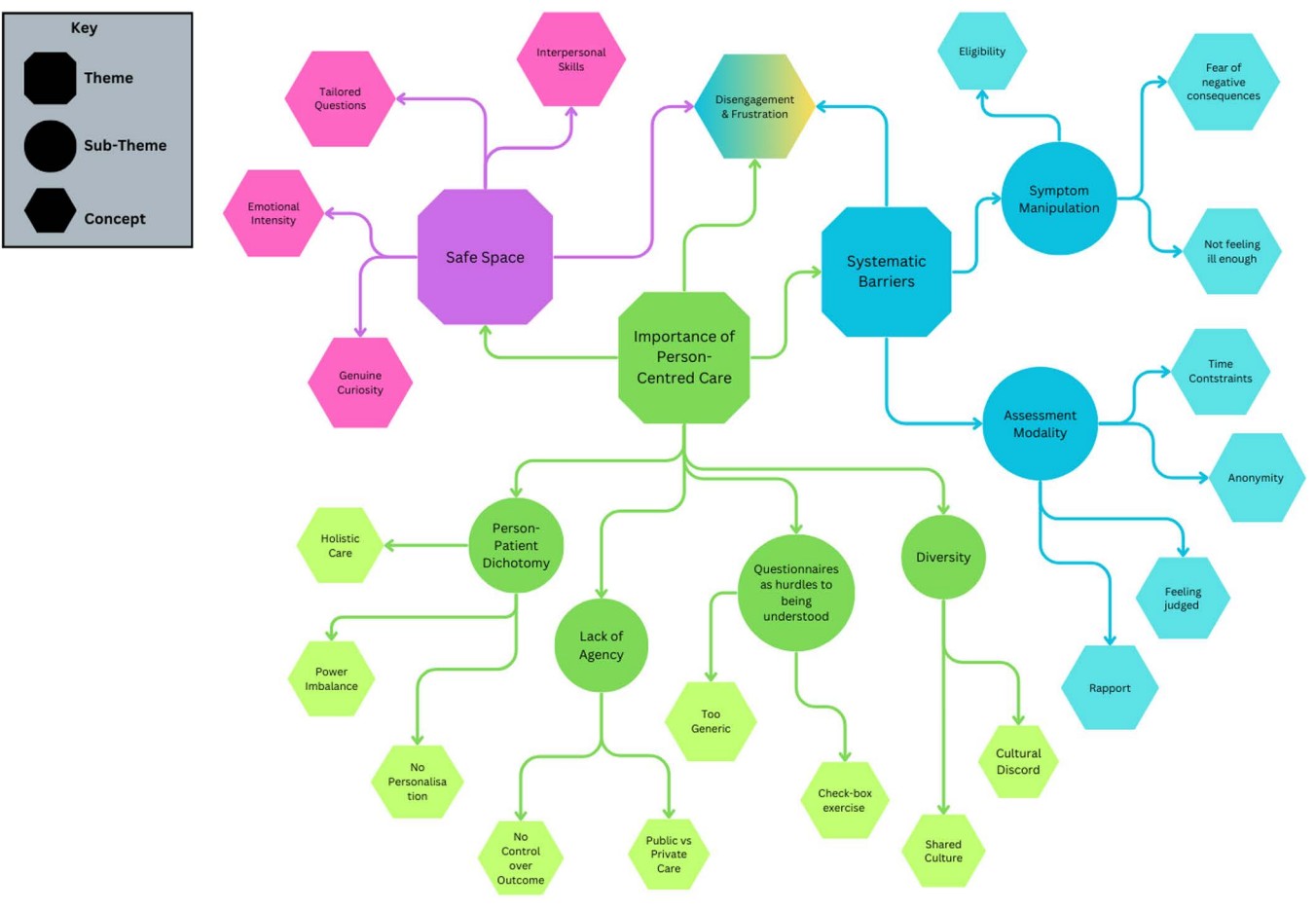

**Fig 2. Mindmap of theme exploration.**

In the presence of these imbalances, participants' felt that their experiences were commonly dismissed and invalidated. This generates frustration and disappointment that led to disengagement and reluctance to attend any subsequent assessments or treatment sessions altogether.

*"… they downgraded my self-harm…saying it was superficial…Downgrading my feelings…And obviously that really upset me. I will reject every assessment from now on." (YP005, female)*

A minority of participants reported a contrasting experience, in which they have felt empowered, comfortable and understood. This highlights the importance of being seen as a person in creating an environment conducive to self-disclosure.

*"Being treated as a person means seeing you for who you are. Making your opinions matter. Understanding that you as a person, you have a choice. It just makes you feel among. It's inclusive." (YP009, non-binary)*

Ultimately, participants felt that a holistic assessment should "focus on everything [and] doesn't have to describe the exact [mental health] problem". "Everything" encompasses one's "family, hopes, dreams, and fears", whereby "conditions

are just one side" of the person. Crucially, a holistic understanding of a person's mental health does not dismiss notions of mental health conditions but carefully incorporates them into a broader picture.

**Lack of agency.** Participants often did not feel in control of the assessments and just went along with what the assessor wanted to know, impeding their ability to disclose. The assessment process "always pivoted back to the structure of the assessment rather than being directed by the person", exacerbating issues of agency.

One person described the ability to have a choice during the assessment as a "basic human right", since "as humans, we should have choices in life", emphasising the importance of their agency during assessment.

Many participants expressed preference for a collaborative approach whereby they are given meaningful choices. For example, participants frequently expressed the wish to have a say in the direction of the assessment, including the questions that would be asked or the selection of any subsequent treatment.

*"Yeah, it would have helped me to kind of know what options there were maybe to have a say in what options I thought would be better for me…" (YP007, male)*

One participant reported a positive experience where they felt like a "co-creator", who led the session and "set what I wanted to discuss" by getting "to decide how far I wanted to go into the topic". This suggests that agency over assessment structure promotes engagement.

In the absence of any choice and self-direction, participants felt like their experiences were moot and that honest self-disclosure may consequently be impeded as it was perceived to be pointless.

*"I think it's never felt in my control. I feel like it's always been directed by others, so I feel like maybe I've not even been able to consider what's important for me to get across because they've decided what's important..." (YP003, female)*

It has also been suggested that the lack of agency during assessment may curb any subsequent engagement during the treatment. Young adults might embrace a passive role and let things happen to them rather than make them happen, reinforcing the superimposed patient role.

*"And so, I just feel like they'll tell me what to do and I'll just be like "yeah". Like, you know, you're just giving the advice for the sake of it, it doesn't feel like you're an active participant in your own recovery or your own journey." (YP006, male)*

Participants who have had experiences with both public and private mental health services also suggested that agency is a financial feature exclusive to private care.

*"Yeah, I think because ultimately, when you go privately, you're paying for a service, so then you have more of a stake and sense of being able to say if the service is adequate or not...And I guess, yeah, you have the power to choose, to some extent, because of money, etcetera, who you go to and what you want to share...whereas obviously in the NHS you just get put with whoever, wherever." (YP004, non-binary)*

Financial investment is perceived to give participants the power to choose their care pathways and the types of information they disclose. By contributing financially, participants feel as though they have paid for the ability to direct their care, as they are holding a stake in the service. Therefore, agency may be seen as a feature absent in public care. The absence of financial investment may further exacerbate perceived power dichotomies and service disengagement.

**Psychometric Tools as Hurdles to being Understood.** Most participants recounted their disconnect with the standardised psychometric tools and their perceived inability to adequately capture things they were struggling with. When they were

encountered, they presented an obstacle that obfuscated participants' real emotions. For example, one patient mentioned how important their loneliness had been at the time of being assessed, yet the questionnaires did not capture that at all.

*"Actually, if it's like "…my loneliness is extremely high" and then tracking that over time would be more meaningful to me than using the PHQ-9 on me. So, I think just making it a bit more person-centred and like a bit less structured around these generic measures would make it feel more relevant and meaningful to me." (YP001, female).*

Some participants found answering "so many questions about so many feelings and thoughts" overwhelming and not conducive to their already emotionally heightened state, making it difficult to respond accurately. For some participants, quantifying their distress numerically has been challenging and counterintuitive, resulting in responses that may not be representative of their actual feelings, impeding meaningful inferences from the resulting scores.

*"I got given like a form to fill out a questionnaire kind of thing and you had to number how you were feeling, like 1 to 10 on loads of different things... Which didn't really work for me because my mind doesn't work like that, like I can't just put a number on how I'm feeling..." (YP007, male)*

Assessments that were filled with standardised questions would often be described as boring and not engaging to participants, dissuading them from being honest. Such assessments were commonly seen as "check-box exercise(s)", potentially revealing a larger automated system that appears to be uninterested in their unique experiences. As a result, some participants responded with a similar disinterest.

One participant, however, did feel as though the questionnaires presented an opportunity to reflect and allowed him to ask himself questions he had not considered before.

*"I think it was good because it sorts of also made me be self-reflective in the fact that, you know, to question my own thought…certain questions like from the PHQ scale… that you normally wouldn't ask yourself. And so, by reading those questions, it gives you a different perspective." (YP006, male)*

**Diversity.** People with different ethnic backgrounds, neurodiversity or sexual identities frequently reported the importance of an in-depth understanding of their distinct diverse background in making sense of their mental health.

Two participants with autism stressed how important an assessor who preferably have lived experience of autism or, alternatively, have experience supporting young people with autism is to understand the "full picture of what my life's like".

*"What I really wish happened, especially with my last assessment, was that they worked with the people at the autism service so that, you know, they're working together." (YP007, male)*

For these people, autism affected their ability to self-disclose and talk about their issues "straight away", especially in a time-sensitive manner. To them, it was important to establish a relationship to the assessor before comfortably self-disclosing.

Achieving a "mutual understanding of what autism is like" and receiving personalised care that respects their differences lays the foundation for a trusting relationship, encouraging honest self-disclosure. Similarly, for participants who were assessed by people from different ethnic backgrounds or sexual orientations, the cultural mismatch generated mistrust and resulted in less self-disclosure due to a perceived fundamental inability to understand their unique experiences.

*"No, I didn't say everything I needed to say. I didn't feel that way. Our sexual orientation and ethnic background weren't common, so I didn't feel the need to." (YP010, female)*

PLOS Mental Health

Several other participants shared that they would have felt more comfortable disclosing their issues, especially those relating to their cultural realities, if they had been paired with an assessor from a similar cultural background. Some suggested that this would enable them find common ground, foregoing any gratuitous explanations or unwanted misunderstandings because the assessor "already understood that aspect of this (issue)", expediting the creation of trust.

When participants felt high perceived prejudice and were aware of stereotypes surrounding their culture, they tried to act in way that minimised stereotype confirmation and any potential harm to them, including being perceived as too highly mentally distressed.

*"Because I know, definitely, being a person with a black skin colour, in this part of the world, has really done a lot of harm to me, and I've had a lot of experiences. So, I didn't want to feel discriminated in any way. I didn't want anyone to maybe generalise the statement, saying, "Oh, that's how the blacks behave," … So, I was just trying my best to avoid such a situation." (YP011, male)*

**Theme 2: Systematic Barriers**

This theme includes details of systematic hurdles that participants encountered, which impeded their ability to self-disclose or elicited negative feelings about assessments. This theme is divided into *assessment modality* and *symptom manipulation*.

**Assessment modality.** Throughout the interviews, participants pointed out distinct trade-offs between face-to-face and telephone assessments. For participants, the former has frequently been associated with the ability to generate strong rapport and facilitating deeper understanding of their emotions via the help of non-verbal cues. Participants felt as though the assessor's ability to see their body language would help them reach a holistic understanding of their issues, wherein their problems are embodied and grounded in observable behaviour and given context.

*"I have personally communicated well through my non-verbal cues…it's also good for them to observe things like how I breathe, things like the pace that I talk, things like my posture. There's a lot that I know can be taken in from a face-to-face thing." (YP009, non-binary).*

Not only was it deemed important for the assessor to observe the help-seeking person, but participants also wanted to see who they were talking to, to form a mutually trusting relationship. This is especially important for people who might have difficulties inferring mood or tone exclusively from auditory cues, wherein face-to face assessments aid in understanding the other person's intentions via non-verbal cues. This perceived reciprocal trust provided an opportunity to disclose things that would have otherwise been left unsaid.

*"…and maybe, not seeing her facially, I would have not opened up to some certain points that she had to question me." (YP010, female)*

However, some participants did report issues with face-to-face assessments whereby their ability to self-disclose was impacted by feelings of being judged negatively. This was emphasised when participants discussed any potential risk of self-harm or suicidal ideation, fearing that a face-to-face disclosure may lead to unwanted consequences.

*"I found in face-to-face sometimes you might have been prone to social desirability. You might not be as truthful… when someone asked you questions about self-harming, you'd obviously say no because you don't want to go to A&E" (YP006, male)*

Additionally, when face-to-face assessments were held in person, participants encountered logistical challenges of arranging and having to pay for the commute to the assessment site. A few participants also reported that their physical disabilities were unaccounted for when making in-person assessment appointments.

*"To deal with the cost of having to go there, using a lift or an Uber, that was difficult." (YP011, male)*

Phone assessments, on the other hand, did not afford a similar safe space, instead participants felt they created an instantaneous distance between them and the assessor, creating a space that was described as "uncomfortable" and "cold". Being embedded in such a space invoked feelings of mistrust and increased some participants' feelings of vulnerability. Furthermore, phone assessments may also exacerbate pre-existing issues of anxiety that may contribute to feelings of discomfort.

*"And a stranger over the phone is not always like the nicest thing. I'm quite a phone-anxious person as well…but I think it does just bring up some feelings of uncertainty, I guess. Anxiety." (YP001, female)*

Conversely, this distance also instilled a sense of anonymity and safety for some, alleviating concerns of being judged. Phone assessments bestowed participants with a sense of control, whereby they could just hang up and leave any potential uncomfortable situation via "the press of a button", bolstering their sense of control.

*"And I guess there's the option of feeling like I could leave because ultimately, I could just hang up the phone. Whereas in person I couldn't like leave as easily," (YP004, non-binary)*

Participants reported both types of assessment, especially phone assessments at the GP and face-to face assessments with psychiatrists, felt like "being rushed along", which consequently diminished perceived opportunities for self-disclosure.

Crucially, both modalities seemed to have perceived benefits and limitations depending on the participant's preferences. Most participants, however, did not get to choose the assessment modality, highlighting previously discussed issues surrounding the lack of personalised care and agency.

*"I would have preferred to have it online, over the phone, or over Teams or Zoom. But then that wasn't the case. It was part of the recommendation that I come in person." (YP010, female)*

**Symptom manipulation.** In addition to feeling as though questionnaires did not accurately capture their moods, some participants were also acutely aware of questionnaires' systematic role in determining service eligibility. When perceiving questionnaires as the barrier to entry, participants have reported thoughts of "not being unwell enough". These feelings may put young people off trying to access help. In light of questionnaires' perceived inability to capture their emotions correctly and a fear of being deemed ineligible, some participants felt the need to "emphasise on the severity of (their) problems" to clearly communicate their distress. A few participants even went so far as to suggest that routine questionnaires would be "very easy to manipulate". As such, they felt encouraged to adjust their symptom scores according to what they believed to be the ceiling for eligibility.

*"I just kind of made them up like I didn't. It was just I put kind of 10 on everyone…I felt like if I put a lower score I wouldn't qualify for the support." (YP007, male)*

As an arbiter of eligibility, the routine deployment of standardised tools was seen as a symptom of the system's disinterest in people's mental health. Thus, participants felt encouraged to "exaggerate" symptoms to ensure that they were "understood properly". In that way, symptom manipulation served to advocate for their best interest.

Disclosure of risk created the inverse phenomenon, whereby participants consciously downplayed symptoms to avoid any consequences that may infringe on their freedom.

*"I was lying through my teeth because if I told them the truth, then I knew that they would just treat me like any other person, and just admit me. So, I couldn't express how things really were..." (YP003, female)*

In either case, the awareness of routine measures' and risk assessments' pivotal role in determining service eligibility diminished people's ability to be truthful about their own experience. Participants instead frequently conformed to what they perceived to be the correct way of being "well" or "unwell" to meet or dodge eligibility criteria.

**Theme 3: Safe Space**

Since assessments hinge on communication between two parties, participants frequently mentioned the importance of the assessor's ability to create a safe space in which participants felt comfortable enough to share details about their struggles that were perceived as embarrassing or otherwise difficult to admit. Participants commonly reported that the creation of such a space begins by having the assessor be open and welcoming, instilling a sense of comfort and acceptance.

*"She just said, "Oh, [name of participant], you're welcome. Can you make yourself comfortable? Feel free to do whatever you want to do. Feel free to do what makes you happy here." And, yeah, that felt welcoming." (YP010, female)*

Generally, participants associated a "warm and understanding" assessor attitude with a safe space. The process of being actively listened to greatly contributed to feelings of warmth and comfort. For instance, participants enjoyed when the assessor reflected their issues back to them or when things were paraphrased, encouraging a mutual exchange of information.

*"But if it's all of the feedback to you, when they paraphrase and sort of including the specificities of your situation, it often makes you feel like they are listening…" (YP006, male)*

Participants almost unanimously stressed the preference for open questions that explore people's mental health rather than closed questions that leave little room for self-expression.

*"And rather than asking specific questions like "do you feel you are at risk?" or whatever he just said, "tell me about you know what's going on. Just tell me everything you know". …He wanted to hear me out and hear everything that I thought was relevant." (YP003, female)*

Similarly, participants wanted the assessor to embrace a genuinely curious stance instead of following predetermined questions. Participants felt this could be accomplished by asking questions tailored to the person's circumstances or asking for clarification when necessary to achieve a mutual understanding of the situation. Ideally, this would culminate in the emergence of a "natural conversation" that transcends the patient-doctor relationships, invoking a sense of familiarity as denoted by wanting to feel as though "I am talking to my friend".

*"It looked like they were more focused on listening than just making notes. Because I think that they asked follow-up questions, and they were nodding the entire time." (YP002, female)*

Participants felt as though assessments could frequently be emotionally "intense", likening them to "spilling (one's) guts" or "pouring (one's) heart out". A warm approach to listening was seen as favourable in attenuating the emotional

strain incurred by the assessment. A lack of warmth, however, compounded feelings of emotional intensity and made it difficult for participants to be honest and vulnerable.

This desired warmth, and safe space were frequently missing from participants' assessments, setting an unwelcoming precedent that negatively impacted the relationship between the assessor and help-seeker, creating a "cold" space instead. Setting such a precedent decreased participants' trust in the assessor and consequently diminished their ability to self-disclose.

For many participants, assessments in which they were actively listened to appeared to be a rarity, emphasising the ubiquity of assessments that lack warmth or failed to establish a safe space. Although "being listened to" was the most common expectation that participants mentioned when seeking assessment, it appears as though that hope rarely materialised.

*"This assessment that just happened shocked me. Because it was nothing like I ever experienced before. The first time I was ever listened to." (YP003, female)*

## Discussion

To our knowledge, the findings of this study provide one of the first explorations of young adults' experiences during mental health assessment. By way of qualitative investigation, we identified several key themes, each pointing to areas of improvement in current mental health assessment practice and elucidating key factors of importance for young adults.

Overall, the results highlight young people's dissatisfaction with mental health assessments. Issues raised primarily revolve around a strong unfulfilled desire for personalised care, involving a lack of agency, the inadequacy of questionnaires, systematic issues with assessments and a lack of interpersonal care. Each of these findings is discussed below in relation to existing literature.

Barring two participants, concerns around the scarcity of personalised care practices were abundant and permeated the accounts of almost every participant. For many, it felt as though assessments prioritised fitting their issues into mental health questionnaires, sidelining their personal stories in favour of the language of psychiatric diagnosis. This invokes feelings of disinterest and subsequently begets disengagement from the service. Evidence suggests that the overemphasis on diagnostic criteria can result in the dismissal of factors such as peer-relationships or socio-economic issues [36], both of which are incredibly valuable for understanding young people's mental health [37,38]. Without obtaining a holistic picture of people's mental health, clinicians may be selecting treatments that are ill-fitting and do not provide the desired outcome [39]. Holistic care could be achieved via the biopsychosocial model of mental health, which emphasises the interactions of biological, psychological and social factors in begetting mental health issues [40]. While this model has found wide academic endorsement [41,42] and resonated with participants' desires, young people did not always receive support provided in line with this model. Importantly, clinicians are constantly under pressure to deliver high quality interventions whilst juggling a high quantity of patients. This dilemma, and the systematic pressure under which clinicians operate in the UK, may impact the ability of clinicians to prioritise more time-intensive, personalised assessments [43]. This misalignment of clinicians' and patients' goals creates an ostensible power imbalance that makes self-disclosure harder. Previous research mirrors this finding, whereby the hierarchical nature of clinical settings skews agency towards the clinician, leaving patients feeling disempowered [44,45]. Consequently, people may find themselves feeling disinterested in engaging as they perceive themselves to hold little power over the outcome of their recovery process [46]. Crucially, participants felt that had they been given more agency by having direct control over aspects of the assessment; for instance, being able to choose the assessment modality or pick a consequent treatment, they would have been more engaged. Increasing people's sense of control during assessment by adopting a collaborative approach that facilitates shared decision making may thus engage people more and subsequently reduce attrition [47,48]. This sentiment resonates with Lines et al.'s [49]

idea of person-directed care. An advancement of person-centred care, person-directed care not only respects the person as a holistic entity beyond the medical context but actively encourages the person to take control of and make meaningful decisions about their care plan, empowering the person to be part of their recovery.

The lack of personalisation further extends into the lack of sufficient assessment accommodations, may that be accommodating people's neurodivergence, sexual orientation or ethnic background. For instance, participants often felt as though their cultural background has not been accounted for and accommodated, resulting in a disconnect between assessor and participant. Accommodating those who might be part of a minority by matching people with assessors from similar cultural backgrounds, as suggested by participants in this study, may remedy these issues. This is especially important because it has been shown that these populations are at risk of falling through the gaps and disengaging due to systemic biases [50,51].

Although participants felt that face-to-face assessments fostered a safe space and trusting relationship, their benefits were not universal. Some participants instead preferred remote or telephone options. Despite these preferences, only very few had a say in the assessment modality. As such, services could meaningfully increase perceived agency by letting patients choose the assessment modality,. Additionally, to rectify issues of power imbalance, clinicians could adopt an intellectually humble stance that deems the patients' knowledge and experience of their own mental health issues as important as one's own clinical appraisal, focusing on co-production as a result [52,53].

Similarly, the present findings suggest that the emphasis and overreliance on standardised questionnaires may alienate people from engaging due to an apparent discrepancy between what was recorded on the questionnaires and the lived experiences of young people. Although these tools are empirically validated [54,55], they seem to insufficiently capture the entirety of people's mental health problems. This becomes especially apparent when considering participants' experience of deliberately altering questionnaires scores because of their perceived power in determining treatment eligibility, posing an obstacle to receiving support. Alternatively, during states of emotional unrest, some participants found these questionnaires can be confusing and overwhelming, resulting in similarly unreliable results. Therefore, participants wished for questionnaires to be supplemented by a variety of open questions that aim to understand the nuances of people's symptoms. Open questioning styles in mental health settings have been shown to yield more detailed and nuanced information of patients' issues, lending themselves to obtaining a holistic picture [56,57].

Nonetheless, there were instances in which participants found assessment(s) to be successful, whereby they provided a safe space that promoted self-reflection and embodied the beginning of people's journeys to get better. In those cases, participants' felt that assessment could provide a deep investigation of one's symptoms as part of their lives, predicated on a safe space that had been created by the assessor. Relying on warmth, active listening and empathy abated uncertainties of self-disclosure and feelings of vulnerability, paving the way for mutual exchange. This sentiment resonates with literature suggesting that such interpersonal skills are fundamental in sustaining the therapeutic alliance and primes patients for beneficial therapy outcomes [58]. Although therapeutic alliance is often talked about in the context of therapy [59], it may be wise to start building this alliance during assessment to facilitate subsequent engagement.

While we did not differentiate between service modalities and can thus not make service specific recommendations, we can make some overarching suggestions on how to improve mental health assessments for young people. Rather than focusing only on improving training for practitioners, implementing service-wide changes may lead to systemic practice improvements. Hence, services may wish to allow for additional time in assessments for patients to express themselves, removing the focus from structured assessments and quantification of symptoms. This does not imply that symptom questionnaires are irrelevant, but they should supplement, and not dictate, personal narratives. Additionally, services could give people meaningful choices about their assessments and treatments. For instance, people could make collaborative decisions about their assessment and treatment modality. This may empower people to feel meaningfully involved in the assessment process

## Strengths & limitations

One strength of this project was that it recruited a diverse group of people in terms of gender, sexuality and ethnicity and obtained rich data. Nonetheless, this project did not specifically recruit people from a disadvantaged socioeconomic background, which may further limit assessment access and beget negative experiences.

PPI was involved as much as possible and contributed to the depth of data this project gathered. Although the lived experience involvement is a strength of this study, it would have been preferable to increase the involvement of our PPI group, for example by involving them in coding transcripts. However, due to limited financial resources, this was not feasible.

This study aimed to broadly explore people's experiences of mental health assessment and thus included people with experiences of various assessment types. However, the UK's mental health service landscape is varied and nuanced, including psychiatric assessments, Talking Therapies assessments, mental health nurses' assessment or GP assessments. All of these may differ in nuanced and meaningful ways, and these differences may have not been captured in this study. Additionally, this study exclusively focused on capturing participants' experiences of assessment, meaning we are unable to draw conclusions about the effectiveness of assessments in allocating people to the "correct" treatment, or for treatment outcomes following support. To ascertain the outcomes of assessment as relating to treatment, future studies should incorporate longitudinal designs, following participants through into treatment and compare assessment experiences with treatment outcomes

Because of this project's financial and temporal constraints, any young person that underwent assessment within the last 18 months was deemed eligible. Given this potentially large gap between assessment and interview, some participants may have had issues accurately recalling their experiences. Moreover, even though this project's remit is defined within the context of young people, due to feasibility issues, we did not recruit people below the age of 18.

Finally, despite the availability of face-to-face interviews, it was advertised to be primarily conducted online. Therefore, some people who might have had issues with remote interviews may have been disinclined to participate.

## Future implications

The results of this study indicate a discrepancy between what young adults require from assessments and what they receive. It is important for mental health assessments to focus on patient-centred care that prioritises idiosyncratic and multifaceted expressions of mental health over standardised methods alone. As suggested by a participant, standardised forms could be filled in collaboratively with the assessor, who may elicit personal narratives of symptomatology alongside standardises scores. Future research should focus on people who have recently undergone mental health assessment to reduce recall biases. It could also be useful to differentiate between and focus on different types of assessments within specific services (e.g., primary care, child and adolescent mental health services). As we suggested in this study, cultural miscommunication may deter any further engagement, thus focusing on minoritised groups of people and investigating their experiences could be useful.

Lastly, future research without temporal or financial limitations should aim to include young people aged below 18 years as there are documented issues with mental health service transitions, which may lead to dissatisfaction and disengagement [15].

## Conclusion

These findings suggest that mental health assessments currently fail in meeting people's demands for holistic care. Currently, participants perceived assessments as largely depersonalised hurdles that are limited by systematic communication barriers and an absence of agency. In extreme cases, this can lead to complete service disengagement, possibly resulting in an escalation of health issues. Yet, assessments appear to have the potential to create the foundation upon which people can start their mental health journeys, facilitating self-reflection and growth. To fulfil that potential and meet

people's demands, new assessment care practices should prioritise person-centred, or even person-directed care, where possible.

## Supporting information

**S1 File. Topic Guide.**
(DOCX)

## Author contributions

**Conceptualization:** Alessandro Dorata, Rebecca Appleton.

**Data curation:** Alessandro Dorata.

**Formal analysis:** Alessandro Dorata, Hannah Andersen.

**Funding acquisition:** Rebecca Appleton.

**Investigation:** Alessandro Dorata.

**Methodology:** Alessandro Dorata, Sarah Bisp, Rebecca Appleton.

**Project administration:** Alessandro Dorata, Rebecca Appleton.

**Resources:** Rebecca Appleton.

**Software:** Alessandro Dorata.

**Supervision:** Sarah Bisp, Rebecca Appleton.

**Validation:** Alessandro Dorata.

**Visualization:** Alessandro Dorata.

**Writing – original draft:** Alessandro Dorata.

**Writing – review & editing:** Alessandro Dorata, Sarah Bisp, Rebecca Appleton.

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
