## [Decision Letter · Decision Letter 0]

6 May 2025

PMEN-D-25-00083

"I wish they heard my story rather than my conditions." – A qualitative exploration of young people's experiences during mental health assessment in the UK.

PLOS Mental Health

Dear Dr. Dorata,

Thank you for submitting your manuscript to PLOS Mental Health. After careful consideration, we feel that it has merit but does not fully meet PLOS Mental Health’s publication criteria as it currently stands. Therefore, we invite you to submit a revised version of the manuscript that addresses the points raised during the review process.

We look forward to receiving your revised manuscript.

Kind regards,

Kaaren R Mathias, PhD

Academic Editor

PLOS Mental Health

Journal Requirements:

1. In the ethics statement in the Methods, you have specified that verbal consent was obtained. Please provide additional details regarding how this consent was documented and witnessed, and state whether this was approved by the IRB.

Additional Editor Comments (if provided):

1. Thank you for your submission and work on this article. At the moment it is not in a form that is suitable for publication and all three reviewers have recommended minor revisions.

2. Overall, it is an insightful study that engages with young people and their access to health care. It uses a well thought out methodology and engages with people with lived experience in the research process. There are some concepts in this paper that need revising and greater clarity e.g. the conclusions suggest the idea that the medical model is separate to person-centred care – I suggest that the authors do more reading and across the paper clarify the definitions and critiques.

a. Zhao, J., Gao, S., Wang, J., Liu, X., & Hao, Y. (2016). Differentiation between two healthcare concepts: Person-centered and patient-centered care. International Journal of nursing sciences, 3(4), 398-402.

b. Lines, Lisa M., Michael Lepore, and Joshua M. Wiener. "Patient-centered, person-centered, and person-directed care: they are not the same." Medical Care 53.7 (2015): 561-563.

c. Salvador-Carulla, L., & Mezzich, J. E. (2012). Person-centred medicine and mental health. Epidemiology and Psychiatric Sciences, 21(2), 131-137.

3. Please revise the whole manuscript carefully to ensure there are no grammatical errors and paying attention to the language styles used – there are several examples of use of informal English – e.g. p 6 Gotten in touch should be rephrased more formally

4. P30 “for a lot of participants” – This is poor grammar – for many participants.

5. There are also some examples of poor grammar e.g. p 30 his misalignment of clinicians’ goals and patient goals creates and ostensible power imbalance that that makes self-disclosure harder

6. The manuscript is longer than it need to be with some sections using many more words than required to convey meaning. E.g. Strengths and limitations section could be expressed much more compactly.

7. The references total over 70 papers – please edit the paper and citations to a maximum of 50 citations that are most pertinent to support your points

8. Methods

a. Was there any effort to ascertain whether participants were from disadvantaged socioeconomic groups?

9. P35 Please review conclusions to make more moderate claims e.g. ‘assessment are plagued’ could perhaps be stated ‘ are limited by … “

a. Eg. , new assessment care practices need to be conceived that prioritise person-centred care, synthesising both it and the medical model into one

b. Could be said “new assessment care practices should prioritise person-centred care”

10. I appreciated the thoughtfulness and reflexivity in the positionality statements offered, although I wondered whether reflexivity around researcher ethnicity and other markers of social hierarchy could also be reflected on.

11. There are some concepts in this paper that need revising and greater clarity e.g. the conclusions suggest the idea that the medical model is separate to person-centred care – I suggest that the authors do more reading and across the paper clarify the definitions and critiques.

Please review and respond to the reviewer comments.

Yours sincerely,

Editor

Reviewers' comments:

Reviewer's Responses to Questions

**Comments to the Author**

1. Does this manuscript meet PLOS Mental Health’s publication criteria ? Is the manuscript technically sound, and do the data support the conclusions? The manuscript must describe methodologically and ethically rigorous research with conclusions that are appropriately drawn based on the data presented.

Reviewer #1: Yes

Reviewer #2: Yes

Reviewer #3: Yes

2. Has the statistical analysis been performed appropriately and rigorously?

Reviewer #1: Yes

Reviewer #2: N/A

Reviewer #3: N/A

3. Have the authors made all data underlying the findings in their manuscript fully available (please refer to the Data Availability Statement at the start of the manuscript PDF file)?

Reviewer #1: Yes

Reviewer #2: Yes

Reviewer #3: No

4. Is the manuscript presented in an intelligible fashion and written in standard English?

Reviewer #1: No

Reviewer #2: Yes

Reviewer #3: Yes

5. Review Comments to the Author

Reviewer #1: Young people’s mental health presents a significant burden on their quality of life.

Mental health assessment may detect issues early and select appropriate treatments

to prevent a worsening of symptoms. However, current research suggests that rates of

non-attendance for mental health services are high amongst young people, possibly

due to negative assessment experiences. Despite their potential importance, little

research has explored the experiences of young people during mental health

assessments.

This study was co-produced with young people with lived experience. We recruited 12

culturally diverse young people who have had a mental health assessment within the

last 18 months. Semi-structured online interviews were conducted to explore young

people’s experiences during assessment. Data was analysed using thematic analysis.

Themes were identified with the help of a peer-researcher and young people with lived

experience.

Three themes were identified importance of person-centred care, systematic barriers

and safe space. Overall, participants reported an unfulfilled desire for holistic and

personalized care that prioritises their needs over meeting systematic requirements.

Assessment was frequently characterised by a lack of agency and dismissal of

experiences, generating disengagement. Young people who reported a more positive

experience of assessment identified factors including a warm environment that

facilitated engagement and prepared participants for upcoming treatment.

The results suggest that, although mental health assessments have the potential to

detect mental health issues early and facilitate subsequent treatment engagement,

young people often experience them negatively. Frequently, they are perceived as

impersonal and rigid, presenting a barrier to help-seeking. In light of these issues,

there is a need to prioritise the implementation of person-centred care in assessment

practices.

A very clearly strong abstract clearly written above.

A very clearly written concise introduction, there were strong evidence of statistical analysis been performed appropriately and rigorously in figures 1, 2 and 3. The data represented was adding depth and rigour to the research as well as the summary and conclusion was robust relating back to the data.

The manuscript technically sound over all , the data seemed to flow concisely well with conclusions especially when comparing the statistics of males and females experiences in surveys responses.

The data provided as part of the manuscript and supporting information, or deposited to a public repository. For example, in addition to summary statistics, the data points behind means, medians and variance measures should be available. If there are restrictions on publicly sharing data—e.g. participant privacy or use of data from a third party—those must be specified.

Yes data was specified in the third party as they had evidence of number of participants and demographics in table and percentage of them when interviewed for this manuscript.

The grammatical language in submitted articles must be clear, correct, and unambiguous. Any typographical or grammatical errors should be corrected at revision, so please note any specific errors.

Revise the sentence structures and rephrase certain words in paragraphs, avoid Jargon.

Also add one or two objective analysis/summary of results to make more clearer senses of the stastics in Table 1. Breakdown of demographic data of study participants. For example what does this mean in terms of over all mental health in the demographic regions, why has these figures impacted mental health in self assessment in general?

Why are percentages lower in certain demographics than others? Write a small analysis underneath to add to the research.

Expand on the results with a summary to make it clear for readers to digest and understand the figures in the table to be precise and concise,.

Double check spelling, commas, headings are correctly formatted with the paragraphs especially with the personal quotes of participants, remove the lines for example.

Double make sure references are cited properly and are in correct alphabetical order with links to research for works. (Accessed at 12/03/2024) for example.

It is a fascinating read, with strong compelling, research elements and components to the study, I like the authors using robust relevant data, statistics, graphs, info graphics and the use of personal quotes from participants.

Just needs minor revision, amend those suggestions/feedback of improvements.

Learnt lots on the mental health, self assessment of young people, very interesting to read about their experiences.

Once revised with minor changes, good to get published! Well done.

Reviewer #2: Well written manuscript and involving young people with lived experience, strengthens the study's authenticity. However, author should clarify how coding disagreements were resolved, when multiple researchers code the same data, differences in interpretation are common. So, Inter-Coder Reliability Score is recommended. Thank you

Reviewer #3: The manuscript addresses an important topic by exploring young people’s experiences during mental health assessments. The research is co-produced with individuals with lived experience, enhancing the study’s relevance. The work foregrounds themes such as the need for person-centred care, the impact of systematic barriers, and the critical role of creating a safe space for open communication.

Major concerns:

The current definition of "mental health assessment" is overly broad and would benefit from a more detailed description. In practice, mental assessments are conducted at varying levels of specialization by providers with different expertise. For example, an interview conducted by a psychiatrist may evoke different experiences and emotional responses compared to a questionnaire-based assessment administered by a social worker or less specialized care provider. The identified themes seem to imply that the focus of the investigation was primarily on assessments using psychometric tools. If that is indeed the case, the authors should clarify and explicitly specify the types of mental assessments being discussed to better capture these distinctions.

The level at which disengagement occurs is also crucial and warrants discussion. It is important to differentiate whether assessments are conducted for checking eligibility for a specific service, during emergency triage by a nurse, or within a consultant or psychiatrist’s clinic. Additionally, clarification is needed on whether disengagement happens at the first encounter with mental health services or as a result of multi-step, specialized assessments. Furthermore, the level of training that assessors have in administering psychological assessments is a significant factor that could influence the experience of agency and subsequent disengagement. If these aspects were not captured in the data collection, they should at least be discussed as limitations.

Since the study was conducted in the UK, it would be beneficial to provide a brief overview of current practices in mental health care and assessment within the country. Describing the typical pathways (e.g., referrals from primary care, the structure of emergency versus routine assessments in public versus private sectors, and how assessments are integrated into overall care) would help contextualize the findings and clarify whether they can be extrapolated to other settings. Noting any policy or systemic factors unique to the UK could further illuminate the specific challenges faced by young people in this context.

The theoretical framework of the study should be specified in the Methods section. It is important to clarify whether any predetermined theories or conceptual frameworks were employed to contextualize the phenomenon under study and to guide the inductive analysis.

The positionality of the researcher should be more thoroughly described. Given that factors such as race/ethnicity and sexual orientation are relevant to the research, it would be beneficial to clarify how the researchers and interviewers are positioned in relation to these diverse backgrounds.

There appears to be some confusion between the terms "gender" and "sex." Sex generally refers to biological and chromosomal differences, typically classified as male or female, whereas gender is a social, personal, and cultural construct that pertains to an individual’s self-identified identity and may not conform strictly to traditional labels. It is particularly important to use appropriate lexicon and clarify this distinction. I recommend that the authors clearly define both terms and use them consistently throughout the manuscript to avoid conflation.

Minor concerns:

Intro:

- In the second paragraph, the manuscript briefly mentions that young people may be hesitant to disclose sensitive information due to stigma or mistrust. I suggest expanding on this point by providing additional context, such as contrasting the experiences of stigma and mistrust faced by young people with those of older age groups, and referencing relevant literature.

- The introduction describes “young people” as those aged 10–24, yet the methods specify a sample of young adults between 18 and 26. This inconsistency in age definitions should be addressed. I recommend clarifying the age range according to the study’s objective and ensuring consistent terminology across all sections.

Methods:

-It is advisable to add a reference for choosing the specific age bracket (18–26) for young adults.

Results:

-The final paragraph in the "Lack of Agency" subsection introduces an important discussion about differences in the perceived sense of agency between private and public services; however, this argument would benefit from further elaboration and stronger support with direct participant quotes.

-In the “Diversity” subsection, in the first paragraph's first line, the term “sexual identity” should be replaced with “gender identity.” If the intended reference is to aspects of sexuality, “sexual orientation” would be the appropriate term to use.

Discussion:

-In the first paragraph, please insert the missing word "of" between "one" and "the first."

-On page 30, line 23, remove the extra "that."

6. PLOS authors have the option to publish the peer review history of their article (what does this mean? ). If published, this will include your full peer review and any attached files.

**Do you want your identity to be public for this peer review?** For information about this choice, including consent withdrawal, please see our Privacy Policy .

Reviewer #1: **Yes: ** Dr Neelam Jayendra Shah

Reviewer #2: **Yes: ** Charles Ganaprakasam

Reviewer #3: No

---

## [Decision Letter · Decision Letter 1]

30 Jul 2025

PMEN-D-25-00083R1

"I wish they heard my story rather than my conditions." – A qualitative exploration of young people's experiences during mental health assessment in the UK.

PLOS Mental Health

Dear Dr. Dorata,

Thank you for submitting your manuscript to PLOS Mental Health. After careful consideration, we feel that it has merit but does not fully meet PLOS Mental Health’s publication criteria as it currently stands. Therefore, we invite you to submit a revised version of the manuscript that addresses the points raised during the review process.

We request you to make changes to the manuscript as suggested by these current reviewers and please note that we are not requesting further editorial edits.

We look forward to receiving your revised manuscript.

Kind regards,

Kaaren R Mathias, PhD

Academic Editor

PLOS Mental Health

Journal Requirements:

1. In the ethics statement in the Methods, you have specified that verbal consent was obtained. Please provide additional details regarding how this consent was documented and witnessed, and state whether this was approved by the IRB.

Additional Editor Comments (if provided):

Thanks for your comprehensive edits to the manuscript. We now request that you make minor revisions as requested by two reviewers.

Thanks very much,

Handling editor,

Kaaren Mathias

Reviewers' comments:

Reviewer's Responses to Questions

**Comments to the Author**

1. If the authors have adequately addressed your comments raised in a previous round of review and you feel that this manuscript is now acceptable for publication, you may indicate that here to bypass the “Comments to the Author” section, enter your conflict of interest statement in the “Confidential to Editor” section, and submit your "Accept" recommendation.

Reviewer #4: All comments have been addressed

Reviewer #5: (No Response)

2. Does this manuscript meet PLOS Mental Health’s publication criteria ? Is the manuscript technically sound, and do the data support the conclusions? The manuscript must describe methodologically and ethically rigorous research with conclusions that are appropriately drawn based on the data presented.

Reviewer #4: Yes

Reviewer #5: Yes

3. Has the statistical analysis been performed appropriately and rigorously?

Reviewer #4: Yes

Reviewer #5: Yes

4. Have the authors made all data underlying the findings in their manuscript fully available (please refer to the Data Availability Statement at the start of the manuscript PDF file)?

Reviewer #4: Yes

Reviewer #5: Yes

5. Is the manuscript presented in an intelligible fashion and written in standard English?

Reviewer #4: Yes

Reviewer #5: Yes

6. Review Comments to the Author

Reviewer #4: The authors have clearly made thorough revisions in response to each reviewer comment. Where recommendations were not implemented, they provided reasoned justifications (e.g., regarding manuscript length or qualitative design limitations). The revised manuscript appears much more polished, precise and aligned with scholarly expectations.

To further strengthen the rigor of the manuscript, I suggest that the authors address the following weaknesses:

1. Although the manuscript acknowledges different types of assessments (e.g., GP vs. psychiatrist vs. talking therapies), it treats them as one conceptual unit. These assessment modalities may differ significantly in tone, setting, purpose and patient experience. Please clarify if participants were clustered by assessment type, and if not, explain why thematic saturation was still appropriate across modalities.

2. The abstract broadly states the findings (e.g., “lack of agency,” “systematic barriers”) without naming any specific examples to increase engagement and specificity (e.g., about psychometric tools or modality mismatch).

3. The discussion focuses on critique and literature alignment but lacks policy implications or practitioner recommendations. For a paper on service design and assessment practice, this limits real-world applicability. Please add a paragraph suggesting practical changes for assessors, service managers or NHS policy e.g., structured training in person-directed communication or embedding choice in modality.

4. The core measure of assessment success is participant comfort or disclosure but no triangulation with service effectiveness or outcomes. While appropriate for qualitative work, this framing might be read as equating emotional comfort with assessment quality. Please add a short paragraph acknowledging that this study focused on experiential validity and future work could compare these insights with clinical outcomes or follow-through into treatment.

Reviewer #5: Minor comments regarding:

a) Section 'PPI Involvement' requires more detail.

b) Findings: the 'Diversity' section seems to imply that two participants and one other share this view. However, is this theme also shared by the rest of the participants, or just by three? Perhaps more clarity here.

c) References section whereby nos. 6 and 21 seem incomplete.

7. PLOS authors have the option to publish the peer review history of their article (what does this mean? ). If published, this will include your full peer review and any attached files.

**Do you want your identity to be public for this peer review?** For information about this choice, including consent withdrawal, please see our Privacy Policy .

Reviewer #4: No

Reviewer #5: No

---

## [Editor Report · Decision Letter 2]

28 Aug 2025

"I wish they heard my story rather than my conditions." – A qualitative exploration of young people's experiences during mental health assessment in the UK.

PMEN-D-25-00083R2

Dear Mr Dorata,

We are pleased to inform you that your manuscript '"I wish they heard my story rather than my conditions." – A qualitative exploration of young people's experiences during mental health assessment in the UK.' has been provisionally accepted for publication in PLOS Mental Health.

Best regards,

Kaaren R Mathias, PhD

Academic Editor

PLOS Mental Health

Thank you for your responses to further reviewer comments. The changes made strengthen the paper and address reviewer concerns. My remaining request is for you to review and edit the paper throughout to move from passive to active tense, particularly in the Methods, Discussion and Implications sections. You can use first person plural to give a more active sense. For example, you say Although this research has not differentiated between service modalities - you could change this to: "While we did not differentiate between service modalities.... "

and instead of 'it may be more fruitful' - you could say 'implementing service-wide changes may lead to changes in practice systemically'

Following language editing, I am happy to proceed with publication of your paper - congratulations!

thanks,

Dr K. R. Mathias